# Physical Education Classes and Responsibility: The Importance of Being Responsible in Motivational and Psychosocial Variables

**DOI:** 10.3390/ijerph191610394

**Published:** 2022-08-20

**Authors:** David Manzano-Sánchez

**Affiliations:** Faculty of Sport Sciences, University of Murcia, Santiago de la Rivera S/N, 30720 Murcia, Spain; david.manzano@um.es

**Keywords:** motivation, violence, school climate, teaching, sex differences

## Abstract

The objective of this research work was to analyse the different profiles that can be identified, based on levels of responsibility in relation to Self-Determination Theory, school climate and violence in Physical Education classes. For this, a total of 470 students of Compulsory Secondary Education or Baccalaureate were given a questionnaire where aspects related to motivation, basic psychological needs, responsibility, school social climate and violence were analysed. An analysis of these profiles was conducted, taking into account the variables of “personal responsibility” and “social responsibility”, with the results leading to the conclusion that three profiles exist: “low responsibility” (n = 89), “moderate responsibility” (n = 187) and “high responsibility” (n = 194). The results reflected statistically significant differences in all the variables between the three profiles. The high responsibility cluster obtained significantly higher values for the different constructs of motivation (except in external regulation, where there were no differences, compared to the “moderate responsibility” group), basic psychological needs and school social climate. On the other hand, it obtained lower values in amotivation and violence, with no differences based on the sex or age of the participants in the distribution of the clusters. It is concluded that the more responsible profile can have positive results in psychological variables in Physical Education classes and in the general educational field. For this reason, the use of active methodologies, which have been extensively studied to promote responsibility in PE classes, could be an appropriate strategy to achieve a more adaptive psychological profile regardless of the gender or age of the students.

## 1. Introduction

Physical education (PE) is considered to provide continuous learning that takes place in the school’s timetabled curriculum and is an entitlement for all learners [1]. This subject is usually a compulsory subject in primary and secondary schools, in which the curriculum has contents about well-being, socio-cultural aspects or psychology [2]. Traditionally, PE settings have focused on sport-specific skills and competition [3]; however, these foci may limit students’ development of personal, social and emotional skills [4]. A fundamental aspect within PE is to look for those aspects that can help improve the effectiveness of teaching [5], and work towards integrating lifelong physical activity by valuing its role in health and citizen coexistence [6].

One of the key factors in human development is responsibility, understood as the ability of people to deal with the consequences of their own decisions [7]. It is important to understand PE as a multidisciplinary area, in which the motivational and relational aspects are fundamental to the generation of a positive attitude in students. In this sense, analysing their responsibility in personal homework can aid the generation of reflection and processes of change [8]. Moreover, exercising social responsibility in this stage of life means ensuring the care and safety of others in the community [9].

High levels of personal and social responsibility in schoolchildren can indicate prosocial behaviours, related to an improvement in psychosocial aspects [10], and in which responsibility shows itself to be a fundamental element, not only for young people who are in a situation of possible social exclusion [11], but also in contexts of a high and medium socioeconomic level [12]. In this regard, studies have demonstrated how responsibility has positive relationships with intrinsic motivation [13,14], causal relationships with autonomy and intrinsic motivation [15,16], self-concept and intrinsic motivation [17], the capacity for learning organisation [9], social climate of the school and the teaching staff [18,19] or a reduction in perceived violence [20].

In connection with this, Deci and Ryan’s Self-Determination Theory (SDT) [21] should be highlighted. These authors indicate that this theory offers a theoretical framework of great importance for educational centres. It establishes that people have three basic psychological needs that are essential and inherent; autonomy (ability to perform actions by one’s own volition), competence (when a person feels able to perform an action) and relatedness (in relation to social relationships) which promote optimal functioning [22], but whose frustration can lead to maladjustment [23]. These play an important role in job efficiency and provide the necessary conditions for psychological growth [24]. It is noteworthy that at the same time, intrinsic motivation itself is positively related to the satisfaction of these basic psychological needs [16].

Classroom social climate is a multidimensional construct [25] that is defined as the way in which both students and teachers perceive the quality of their experiences in the classroom [19]. This variable stimulates learning by improving student motivation, basic psychological needs and academic performance [26]. In the same way, violence is defined as an intentional conduct through which damage or harm is caused [27]. Said violence is currently presented as a problem of great impact in the field of Compulsory Secondary Education and, therefore, requires an effective and immediate multidisciplinary approach [28], relating the variables of the classroom social climate with a reduction of violence [29].

Although there are several studies in the scientific literature that address profiling based on motivation levels [18,30], there are no studies that have used responsibility in order to create profiles and relate them with socially adaptive aspects, such as violence and the school social climate, while applying the SDT theory in the context of PE [22].

This study contributes to the literature in two important ways. First, we test whether the levels of responsibility of PE students could result in different values in some of the most studied variables in the field of PE (motivation, basic psychological needs) and in the educational field in general (school social climate and violence). Secondly, a fundamental aspect is to observe whether there are differences depending on the age and gender of the participants in the profiles carried out, given that in the bibliography cited, the results are not entirely illuminating in this respect and neither with regard to educational level. Studies can be found indicating higher levels of responsibility in girls [31] and others in boys [32], while others claim the opposite [33].

Therefore, the main objective of this study is to carry out a profile analysis, verifying the levels of responsibility of PE students of a Compulsory Secondary Education and Baccalaureate. At the same time, the intention is to observe whether there are differences in levels of autonomy, competence and social relationship, as well as those of motivation, school social climate and perceived violence, in connection with levels of responsibility. The third and last objective will be to check whether these profiles present differences in the sex or age of the participants.

## 2. Materials and Methods

### 2.1. Study Design and Participants

This research was a quantitative transversal and descriptive study. We requested informed consents (confidential data treatment and participation in the study) from the students and their parents (in case of under 18 participants). This document was attached with a letter of introduction to the schools and the approval of the ethics committee of the University of Murcia (1685/2017).

Participants were selected from three different public secondary schools based on accessibility and convenience. The sample of students originally consisted of 501 participants. The following exclusion criteria were established: (a) to complete all test scales, (b) to complete the pre- and post-tests on both occasions, and (c) to complete at least 90% of the test items (excluding double answers). After applying the exclusion criteria and calculating Mahalanobis distance to remove outliers, the final sample consisted of 470 participants with ages from 11 to 20 years old (M = 14.77 SD = 2.01), of which 272 were boys (57.9%) and 198 were girls (42.1%). Figure 1.

### 2.2. Procedure

First, after obtaining permission from the ethics committee, the centres that participated in this study were contacted, and an explanatory seminar held with the managers. Once the approval was received from the centre, the tutor teachers were contacted so that the students could be given an informed consent that they had to bring back signed. Afterwards, the study was explained to the PE teachers and the questionnaire, in the form of a Powerpoint presentation, was given to the students. A further explanation to clear up any possible doubts on the completion of the test was also made prior to the test itself, which lasted a total of 20 to 30 min. At the time of handing out the questionnaire, the principal investigator was present, giving instructions to students and trying to create a suitable atmosphere. The variables to be measured in the research were responsibility, autonomy, competence and social relations, levels of motivation, the school social climate and violence. The study was conducted during the 2018–2019 and 2019–2020 academic years, with the centres having a similar socioeconomic context.

### 2.3. Instruments

A closed-question questionnaire was used in the study. It was divided into two parts: the first one dealt with sociodemographic variables, while the second included the different scales used in the study. All scales contained a reliability process in order to ensure their adequacy for students using the omega coefficient, taking into account that Cronbach’s alpha coefficient is affected by the number of items and the number of responder alternatives [34] and works with continuous variables, something that does not occur in the social sciences, which underestimates reliability [35]. The following scales were administered:Personal and Social Responsibility (PSRQ): to measure the personal and social responsibility level. The Spanish validated version of the Personal and Social Responsibility Questionnaire [36] was used with 14 items from two variables (personal and social responsibility). Participants responded on a Likert-type scale from 1 (Totally disagree) to 6 (Totally agree). The instructions were presented at the beginning of the questionnaire along with the following statement: “It is normal to behave well at times and badly at other times. We are interested in finding out how you normally behave in PE classes. There are no correct or incorrect answers. Please answer the following questions, choosing the option which bests represents your behaviour”. Reliability was of personal responsibility (Ω = 0.902) and social responsibility (Ω = 0.915).Psychological Need Satisfaction in Exercise (PNSE): to measure the satisfaction of the need for social competence, autonomy and relationships. The scale was adapted for Spanish and to the education context by Moreno et al. [37]. It is a scale with 18 items (6 from each variable). The variables were autonomy, competence and relatedness. These were preceded by the sentence “During my class…” and the answers were provided on a Likert-type scale ranging from 1 (False) to 6 (True). Reliability was of autonomy (Ω = 0.954), relationship (Ω = 0.956) and competence (Ω = 0.944).Motivation toward Education Scale (in French, EME): to measure the different types of motivation. The Spanish version of the Échelle de Motivation en Éducation [38] validated by Nuñez et al. [39] was used. The questionnaire consists of seven subscales, called intrinsic motivation to know; intrinsic motivation to accomplish; intrinsic motivation to experience sensations; identified regulation; introduced motivation; external motivation; and amotivation. The instrument is composed of 28 items preceded by the sentence “I go to school/high school because…”, with a seven-point Likert-type scale, from 1 (totally disagree) to 7 (totally agree) and distributed into seven subscales, five of them containing four items and two of them containing three items. The Omega values of the different variables were: intrinsic motivation to achievement (Ω = 0.962) intrinsic motivation to experience (Ω = 0.953), intrinsic motivation to knowledge (Ω = 0.961), Identified regulation (Ω = 0.945), Introjected regulation (Ω = 0.954), external regulation (Ω = 0.949), amotivation (Ω 0.960).Questionnaire of School Violence (CUVE): to evaluate violence perception. It was designed by Álvarez et al. [40]. This questionnaire is composed of 41 items in eight different categories, which were adapted to primary and secondary education contexts by Álvarez et al. [41]. Answers are provided on a Likert-type scale ranging from 1 (totally disagree) to 5 (totally agree). In this study, the global scale was used, with a value of Omega’s coefficient of 0.885.Questionnaire to assess school social climate (CECSCE): to evaluate the climate perceived by the students with regard to their class, teacher and school. It was designed by Trianes et al. [42] and validated in a 12–14 years old sample. The questionnaire consists of two subscales called “Climate relative to the school” (e.g., “Students are really willing to learn”), made up of eight items, and “Climate relative to the teaching staff” (e.g., “Teachers of this school are friendly to students”), composed of six items. A five-point Likert-type scale was used, ranging from 1 (totally disagree) to 5 (totally agree). The Omega values of the two scales were of Ω = 0.914, for the school teacher climate, and Ω = 0.90 for the school class climate.

There was an adequate value following Campos-Arias and Oviedo [43] considering acceptable values from 0.70. In this sense, we also checked the Cronbach’s alpha values to ensure an acceptable value, following the criteria of Viladrich et al. [44].

### 2.4. Statistical Analysis

The first step was to analyse the descriptive values of the mean, standard deviation, skewness, and kurtosis, taking into account that values <1.96 are considered as normal in kurtosis and skewness [45]. Moreover, Omega’s coefficient was estimated to examine the reliability of each variable analysed. The second step was to analyse the correlation between variables, considering values <0.80 to check the absence of multicollinearity [46]. In addition, we calculated the sample size, and it was considered adequate for the population, with a confidence level of 95% and error range of 5% (n > 383).

After that, we followed the approach that uses hierarchical and non-hierarchical methods combined [46]. The first step on the process of analysis used the hierarchical procedure to identify the right number of clusters following the dendrogram. We used Ward’s model, standardising the variables using Z scores of personal responsibility and social responsibility. The most suitable solution was selected using squared Euclidean distance and analysing the dendrogram. Through the dendrogram, it was possible to determine a cut distance to define which would be the group formed. However, according to Hair et al. [46], this decision is subjective and should be made with the objective of analysis and the number of desired groups. Following the study of Bussab [47], the cut level of the dendrogram should be conducted by analysing it to find meaningful changes in the levels of similarity between the successive fusions. Regarding the dendrogram above, we cut close to the 8th level, resulting in the identification of three groups. Hence, it was decided to use these three profiles. After which, a nonhierarchical cluster, using the k-means test, was conducted and we made a validation splitting the sample in two sub-samples, finally selecting the same profiles with three possibilities (low responsibility, moderate responsibility and high responsibility).

The final step was to check the differences between profiles according to motivation, basic psychological needs satisfaction, school climate and violence. To do this, a multivariate analysis of variance (MANOVA) was performed with the F-value and size effect (eTa). A post hoc test was used with Bonferroni correction. Furthermore, we checked the differences between profiles, taking into account the age of the sample in the three groups and their sex. All analysis was performed with IBM SPSS, v. 23.0 (IBM, Armonk, NY, USA) establishing the level of significance *p* < 0.05 (*) or *p* < 0.01 (**).

## 3. Results

### 3.1. Descriptive Analysis and Bivariate Correlations

Descriptive statistics (mean, standard deviation, skewness and kurtosis), the Omega values of the subscales, as well as the bivariate correlations for all the variables under the study are presented (see Table 1). The data indicate a very similar score in both types of responsibility (M = 4.92 and M = 4.93), highlighting the highest values of the scale in identified regulation (M = 5.33) and external regulation (M = 5.78) in relation to motivation and a score of 3.98 for relationship (on the basic psychological needs satisfaction scale), teacher climate (3.67 vs. 3.41 for school climate) and a score of 2.34 for violence. The bivariate correlation analysis demonstrated that all the variables had a significant relationship with each other at *p* < 0.05 or *p* < 0.01, excluding violence, which was only negatively correlated with the kinship variable, school climate and teaching climate, as well as positively with amotivation.

### 3.2. Profile Analysis

The cluster analysis was conducted in order to study the profiles of the students based on the levels of personal and social responsibility, in order to subsequently assess how these profiles influence the rest of the variables under study.

To determine the levels of responsibility among the groups, a hierarchical cluster analysis was performed using the method specified in the data analysis section, with the profiles passed through the k-means test as the final statistical method. Finally, three profiles called low responsibility (n = 89; 18.9%), moderate responsibility (n = 187; 39.8%) and high responsibility (n = 194; 43.3%) were developed, as observed in Figure 2 using the Z scores for personal responsibility and social responsibility.

### 3.3. Differential Analysis of Player Profiles According to Levels of Responsibility

To analyse the characteristics of each profile, a multivariate analysis of variance (MANOVA) was performed with the clusters as independent variables and the variables under study as dependents (see Table 2).

The adequacy of the multivariate analysis was verified by using the Box test for the equality of covariance matrices (m = 256.843; *p* < 0.001), considering the use of this type of procedure as adequate. Significant differences were found at the multivariate level (Pillai trace = 0.418, (F (241,710) = 10.071, *p* < 0.001). The subsequent univariate ANOVAs reflected significant differences between all the variables considered; with the differences between groups being described by means of the Bonferroni test.

The results in Table 2 indicate that the “high responsibility” cluster had different and statistically significant values in all the variables compared to the “moderate responsibility” and “low responsibility” cluster. In turn, post hoc analysis reported that there were statistically significant differences between the three groups, with the “low responsibility” cluster having lower values in all variables, except violence. With this variable, statistically significant differences were obtained only with respect to the “high responsibility” cluster. In fact, this cluster had statistically significant values greater than the other two groups in all variables except violence, where they were lower. These differences were significant, with a value of *p* < 0.01, except in terms of the climate of the centre of the “low responsibility” and “moderate responsibility” groups (*p* = 0.013), and in the external regulation between the “high responsibility” and “moderate responsibility” groups (*p* = 0.023).

### 3.4. Differences between the Profiles According to the Sex and Age of the Participants

To verify the differences between the different profiles according to sex and age, it was decided to conduct an analysis using the Pearson’s chi-square statistical technique through cross tables. This test allows you to compare the observed and expected frequencies in each category, to test whether all categories contain the same proportion of values or whether each category contains a user-specified proportion of values. Similarly, the corrected typified residuals provide us with information on where the differences found were used. It is considered that values greater than 1.90 are indicators of dependence between these two categories and that, therefore, the differences would be significant.

Table 3 indicates no difference between the three profiles and sex or age. For this reason, the profiles had a similar distribution when considering the sex and age of the participants. We also checked the correlation between age and personal and social responsibility, and the values were not statistically significant. This allows us to corroborate the results obtained in Table 3, without differentiating the levels of responsibility based on age.

## 4. Discussion

The purpose of this study was to conduct an analysis of profiles based on the levels of responsibility and try to verify their relationship with different incident psychosocial variables in students of Compulsory Secondary Education and Baccalaureate in PE classes.

Regarding the first of the objectives, the three profiles of “low, moderate and high” responsibility were identified. In relation to other studies that have contemplated the analysis of profiles in students, we can indicate that the distribution follows a line of studies such as that of Gómez-López et al. [48], where motivation was studied as a variable to extract the profiles. Although many studies identify four clusters when this variable is analysed [18,49,50,51], others that study motivation distinguish only between low and high motivation [52,53].

The analysis of profiles that consider other variables in the field of non-university education is not very widely documented, highlighting research such as that of da Rocha et al. [54], analysing student engagement, or that of Regueiro et al. [55], valuing learning goals, which is a novel study in the realisation of these profiles in terms of responsibility. In the case of university students, there are several investigations that support the use of the profile technique for variables such as academic success, mental attitude, the value of education for the future or learning style [56,57,58,59]. Regarding the sample size, we also found large differences between the use of samples of less than 200 subjects [58] or more than 2000, as in the study by Inglés et al. [51]. Whatever the circumstances, the number of profiles analysed was generally between two and four, regardless of the variable under study and the educational stage.

In relation to the second of the objectives, seeking to verify the relationship between the responsibility profiles and the variables under study, we highlight that our study verifies how the profile that was called “high responsibility” obtained higher values in aspects related to motivation, with subjects showing more self-determination, greater satisfaction of the three basic psychological needs and lower levels of amotivation in relation to the rest of the variables. The same can be corroborated by different studies, where they have identified a positive relationship between high levels of responsibility and the variables indicated above in Secondary Education [12,18,19,60,61] and Baccalaureate [62,63] students. It is noteworthy that there are several studies that have been analysed in PE classes, such as the promotion of responsibility, which supposes a more adaptive psychological profile in these stages [60,62,63]. This aspect is fundamental if we consider that adherence to school and academic success are highly related to motivation [64], together with the satisfaction of basic psychological needs [65]. In the same file, different studies include these variables in order to achieve greater adherence to physical activity and satisfaction with PE classes [66,67].

At the same time, this profile demonstrated significantly higher values than the other two profiles in the school climate and teacher climate, together forming a school social climate. The scientific literature supports this direct correlation between responsibility and the school social climate, both when predictive models are made [12], and by applying methodologies such as the Personal and Social Responsibility Model or TPSR [68,69]. Correspondingly, there is no scientific literature where non-significant or negative results are appreciated in relation to these variables. It is essential to consider the great importance of promoting the school climate of students and teachers, due to its relationship with student achievement [70] and in PE classes such as in the study of Hyeon et al. [71], with near of 3000 students (control and experimental students) and 42 teachers, whose main conclusion is that autonomy-supportive teaching including variables such as promoting responsibility and autonomy in PE classes can lead to an improvement in the school climate and prosocial behaviours.

Furthermore, violence turned out to be lower in the “high responsibility” group compared to the others. In this sense, we find studies that verify how personal and social responsibility is inversely related to the levels of violence perceived and suffered [12,20,72], with more studies being necessary, especially of intervention. The reason being, there are studies that do not observe differences in the levels of violence after interventions that seek to promote responsibility [12] and others that do verify this aspect, especially with the promotion of social responsibility [20]. It is noteworthy that violence and school climate are also negatively related. Therefore, promoting an adequate school climate is essential to control this aspect, especially in conflictive centres [73]. In PE classes, violence situations are usually related with to problems such as dissatisfaction with one’s body [74] or perception of low physical competence [75]; however, on the other hand, PE is one of the most appreciated classes by students [76]. In this sense, studies such as Valero-Valenzuela et al. [77] are very interesting, given that, just as in our study we found that the most adaptive profile had lower levels of violence, this study reached the same conclusion, adding in turn the importance of a teaching profile that encourages the autonomy of students.

Finally, the third objective sought to observe whether there were differences in the levels of responsibility depending on the age and gender of the participants. This was an objective whose hypothesis was not entirely clear, given that there are studies that observe differences according to gender in the values of responsibility, with higher values in boys [32] and others in girls [31], while others fail to identify such differences [63]. The same occurs with age, where research is observed that identifies higher values of responsibility in younger students [32] and others in older ones [33]. It is essential, therefore, to consider interventions and cross-sectional studies. In this sense, research by Nasaescu et al. [78] should be highlighted, where a longitudinal study was conducted with a group of 9 to 12 and 12 to 16 year olds, finding that in both cases and especially in the older children, antisocial behaviours are maintained over time. This makes it essential to think about working with the last years of elementary school on aspects related to responsibility and prosocial behaviours, when programs that seek to prevent aspects related to disruptive or antisocial behaviour can be applied [12,32,60,63]. Here, it would be especially interesting to use longitudinal studies of more than one academic year [79,80,81]. The aim would be to promote positive experiences in the classroom, which allow a greater adoption of responsibility and the achievement of adaptive behaviours [82].

### Limitations and Future Lines of Research

Regarding the limitations of this study, it should be noted that the sample size, although considered adequate, could have been expanded by using elementary school students, in order to compare the different educational stages. In addition, the use of qualitative instruments, such as interview analysis, could have been considered to corroborate the quantitative results. It is worth highlighting the need to carry out research that seeks to apply interventions in order to promote responsibility and aspects related to SDT, due to the large amount of existing literature that reflects an improvement in numerous variables thanks to the promotion of responsibility and motivation together with the satisfaction of basic psychological needs. This could be especially interesting to PE classes due to their characteristics such as the continuous interactions between the participants, the expectations towards the subject or the proximity relationships between teachers and students [83] and the good expectations to this subject [76].

Future studies should contemplate differentiating according to the sex of the participants, especially in intervention studies, as well as considering the educational stage, socio-economic level and the subject context due to controversial studies regarding whether the characteristics of PE classes can be optimal for an improvement of this variable [84]. Here, the extension of longitudinal studies to all educational levels would clearly be of great interest and apply different methodologies taking into account participant context. Finally, including the families’ perception about their children’s behaviour outside of PE classes and school in relation to responsibility could be interesting.

## 5. Conclusions

It is concluded that there are different profiles depending on the levels of personal and social responsibility in PE classes during the Secondary and High School stage. In turn, these profiles indicate that PE students who had higher levels of responsibility are also those who had more internal motivation and greater satisfaction of basic psychological needs, while generating a better school social climate and lower values of violence.

It is noteworthy that these positive results occur for the profile with more responsibility regardless of age or gender, while no differences in the distribution of profiles based on any of these variables were found.

For these reasons, the promotion of responsibility in PE classes, regardless of the gender or age of the students by teachers might be an important aspect to take into account, as well as values related to this subject, such as effort or respect for others.

In the same point, the use of active methodologies, such as the widely documented TPSR or cooperative learning in PE classes, could be an adequate strategy to achieve this aspect.

## Figures and Tables

**Figure 1 ijerph-19-10394-f001:**
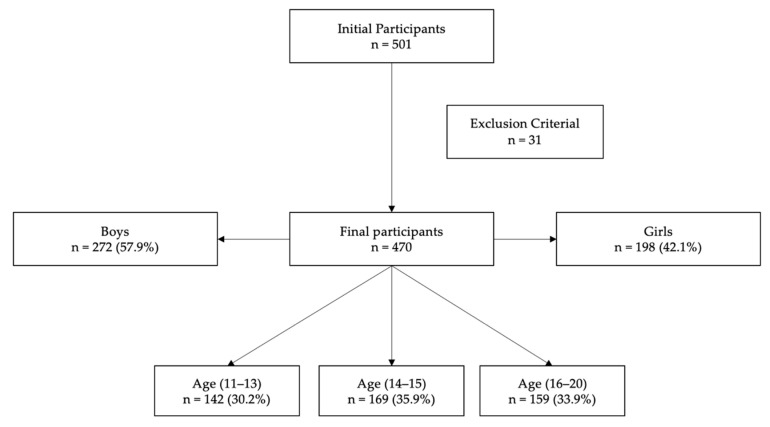
Flow of participants.

**Figure 2 ijerph-19-10394-f002:**
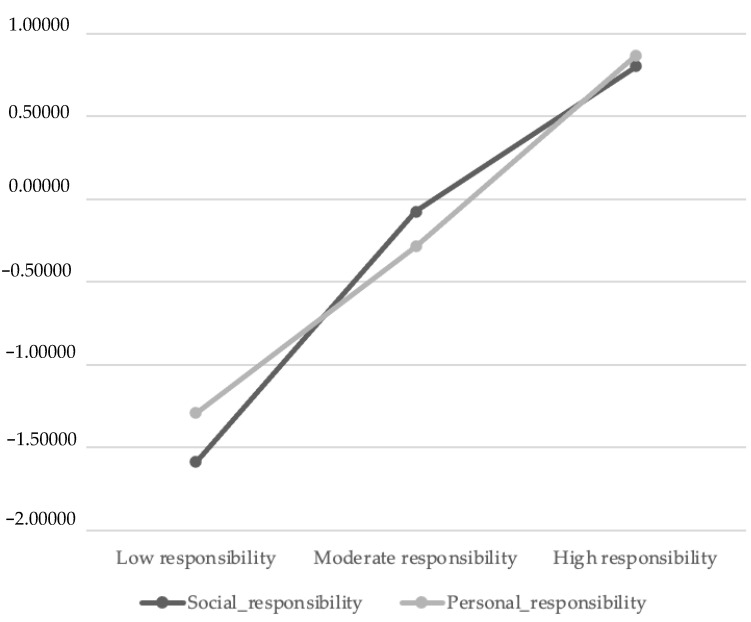
Profile analysis. Differences according to responsibility.

**Table 1 ijerph-19-10394-t001:** Descriptive statistics and bivariate correlations of the sample.

		Range	M	SD	S	K	Ω	1	2	3	4	5	6	7	8	9	10	11	12	13	14	15
1	Autonomy	1–5	3.43	0.87	−0.298	−0.238	0.954	-	0.661 **	0.450 **	0.525 **	0.555 **	0.497 **	358 **	0.491 **	0.284 **	0.017	0.567 **	0.590 **	−0.065	0.442 **	0.409 **
2	Competence	1–5	3.77	0.75	−0.552	0.140	0.956	-	1	0.443 **	0.542 **	0.415 **	0.521 **	0.463 **	0.485 **	0.345 **	−0.077	0.476 **	0.545 **	−0.016	0.539 **	0.503 **
3	Relatedness	1–5	3.98	0.85	−0.939	0.643	0.944	-	-	1	0.317 **	0.374 **	0.310 **	0.291 **	0.313 **	0.277 **	−0.097 *	0.541 **	0.487 **	−0.117 *	0.343 **	0.411 **
4	IM_Knowledge	1–7	4.90	1.31	−0.494	−0.112	0.961	-	-	-	1	0.680 **	0.707 **	0.558 **	0.655 **	0.397 **	−0.057	0.420 **	0.471 **	−0.061	0.476 **	0.410 **
5	IM_Experience	1–7	4.10	1.40	−0.149	−0.675	0.953	-	-	-	-	1	0.557 **	0.379 **	0.558 **	0.293 **	0.019	0.433 **	0.467 **	−0.006	0.355 **	0.312 **
6	IM_Achievement	1–7	4.92	1.37	−0.476	−0.311	0.962	-	-	-	-	-	1	0.516 **	0.702 **	0.430 **	−0.018	0.348 **	0.418 **	0.047	0.480 **	0.374 **
7	Identified R.	1–7	5.33	1.11	−0.551	−0.138	0.945	-	-	-	-	-	-	1	0.477 **	0.590 **	−0.225 **	0.244 **	0.365 **	−0.058	0.381 **	0.375 **
8	Introjected R.	1–7	5.10	1.31	−0.557	−0.253	0.954	-	-	-	-	-	-	-	1	0.449 **	0.057	0.382 **	0.431 **	0.051	0.445 **	0.367 **
9	External R.	1–7	5.78	1.06	−0.819	0.233	0.949	-	-	-	-	-	-	-	-	1	−0.103 *	0.170 **	0.245 **	0.043	0.324 **	0.313 **
10	Amotivation	1–7	2.23	1.45	1.184	0.648	0.960	-	-	-	-	-	-	-	-	-	1	0.001	−0.076	0.199 **	−0.142 **	−0.142 **
11	School climate	1–5	3.41	0.77	−0.165	−0.255	0.900	-	-	-	-	-	-	-	-	-	-	1	0.716 **	−0.202 **	0.349 **	0.439 **
12	Teaching climate	1–5	3.67	0.79	−0.343	−0.487	0.914	-	-	-	-	-	-	-	-	-	-	-	1	−0.136 **	0.439 **	0.443 **
13	Violence	1–5	2.34	0.89	0.484	−0.308	0.885	-	-	-	-	-	-	-	-	-	-	-	-	1	−0.052	−0.138 **
14	Personal:_Responsibility	1–6	4.92	0.85	−0.836	0.086	0.902	-	-	-	-	-	-	-	-	-	-	-	-	-	1	0.650 **
15	Social_Responsibility	1–6	4.93	0.79	−0.627	−0.127	0.915	-	-	-	-	-	-	-	-	-	-	-	-	-	-	1

Note: * *p* < 0.05; ** *p* < 0.01; M = Mean; SD = Standard deviation; S = weakness; K = Kurtosis; Ω = Omega’s Coefficient; IM = Intrinsic Motivation; R = Regulation.

**Table 2 ijerph-19-10394-t002:** Analysis of the profiles according to the perceived motivational climate.

	Low Responsibility		Moderate Responsibility		High Responsibility				
	M	SD	M	SD	M	SD	F	η	*p*
Autonomy	2.78	0.86	3.37	0.75	3.80	0.79	52.173	0.183	**
Competence	3.03	0.80	3.75	0.58	4.14	0.61	91.172	0.281	**
Relatedness	3.46	0.85	3.90	0.80	4.29	0.75	34.930	0.130	**
IM_Knowledge	3.74	1.34	4.88	1.06	5.44	1.18	65.343	0.219	**
IM_Experience	3.34	1.30	3.99	1.25	4.56	1.41	26.980	0.104	**
IM_Achievement	3.72	1.34	4.98	1.14	5.43	1.25	59.587	0.203	**
Identified R.	4.47	1.19	5.32	1.01	5.73	0.94	46.563	0.166	**
Introjected R.	4.04	1.34	5.08	1.14	5.60	1.15	53.117	0.185	**
External R.	5.09	1.18	5.81	1.00	6.08	0.91	29.957	0.114	**
Amotivation	2.68	1.34	2.24	1.44	2.03	1.48	41.091	0.150	**
School climate	2.99	0.70	3.26	0.73	3.75	0.70	57.182	0.197	**
Teaching climate	3.09	0.75	3.57	0.69	4.04	0.72	52.173	0.183	**
Violence	2.39	0.77	2.46	0.85	2.19	0.96	91.172	0.281	*

Note: M = Mean; SD = Standard Deviation; IM = Intrinsic Motivation; R = Regulation; F = Multivariate value. * *p* < 0.05; ** *p* < 0.01; η = eTa square.

**Table 3 ijerph-19-10394-t003:** Differences according to sex and age.

	Low Responsibility			Moderate Responsibility			High Responsibility					
	Total	%	R	Total	%	R	Total	%	R	χ^2^	gl	*p*
Menu	55	20.2%	0.5	105	38.6%	−0.3	112	41.2%	0	0.792	2	0.673
Women	34	17.2%	−0.6	82	41.4%	0.4	82	41.4%	0.1			
11–13	24	16.9%	−0.6	62	43.7%	0.7	56	39.4%	−0.3	3.394	4	0.494
14–15	29	17.2%	−0.5	64	37.9%	−0.4	76	45.0%	−0.7			
16–20	36	22.6%	1.1	61	38.4%	−0.3	62	39.0%	−0.4			

Legend: R = Standardised Residual; SD = Standard Deviation; χ^2^ = chi squared; gl = deegres of freedom.

## Data Availability

Data collected and analysed during the study are available upon reasonable request.

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
