# Peer review of "Physical Education Classes and Responsibility: The Importance of Being Responsible in Motivational and Psychosocial Variables"

_ijerph, 2022, doi:10.3390/ijerph191610394_

Round 1
Reviewer 1 Report
Dear authors
I have enjoyed reading your manuscript “Physical Education classes and responsibility: the importance of being responsible in motivational and psychosocial variables”, which is interesting, rigorous, and adequate. However, I am going to make some suggestions for improvement that I think can contribute to increasing the quality of the manuscript.
a) The introduction is very detailed and precise, with a thorough literature review.
b) Material and Methods. Consider including a figure with the flow of participants, as well as age and gender distribution.
c) The results and discussion seem to me to be very complete and with good statistical analyses. Comparing the results obtained with other related and recent studies.
d) Conclusion, it might be interesting to refer to limitations and future lines of research.
e) Consider including a more explicit mention of the limitations and future prospects of the study.
I encourage you to go deeper into the study and to research the current literature on the subject, which will enhance the quality of your research, which is undoubtedly interesting.
Author Response
Thank you to the reviewer for your work. I display the response to the reviewer in red. In the manuscript, the changes have been reflected with change control and underlined in yellow color
Dear authors
I have enjoyed reading your manuscript “Physical Education classes and responsibility: the importance of being responsible in motivational and psychosocial variables”, which is interesting, rigorous, and adequate. However, I am going to make some suggestions for improvement that I think can contribute to increasing the quality of the manuscript.
a) The introduction is very detailed and precise, with a thorough literature review.
Thank you
b) Material and Methods. Consider including a figure with the flow of participants, as well as age and gender distribution.
A flow of participants figure have been included (figure 1)
c) The results and discussion seem to me to be very complete and with good statistical analyses. Comparing the results obtained with other related and recent studies
Thank you
d) Conclusion, it might be interesting to refer to limitations and future lines of research.
The conclusion have been changed taking into account the limitations and future lines of research
e) Consider including a more explicit mention of the limitations and future prospects of the study.
A limitation and future line section have been included in discussion.
I encourage you to go deeper into the study and to research the current literature on the subject, which will enhance the quality of your research, which is undoubtedly interesting.
The bibliography has been expanded with more references in relation to physical education and the variables under study. In the same way, the discussion and the conclusions have been modified (in yellow) taking into account their considerations and seeking the link with physical education.
Thank you to the reviewer for all your comments, I consider the manuscript has improved substantially thanks to your considerations.
Reviewer 2 Report
Dear Author,
It was a great pleasure to read your article. The topic is very actual and extremely important. And – what the most important – study design and research realization are very satisfactory. My concerns are connected with the conclusions of the study.
Please consider my suggestions, because there are some points to improve.
1. Abstract – please rethink using the statement „three basic psychological needs”. It is not correct.
2. Abstract – „It is concluded that the promotion of responsibility in Physical Education classes and in the general educational field, can have positive results in obtaining a more adaptive psychological profile”. Please notice, that we can not indicate the direction of these relations. Also, the more adaptive psychological profile can have positive results in responsibility in Physical Education classes. We can only diagnose the relations.
3. Introduction – line 40: „homework?”. It must be a mistake
4. Conclusions – „It is concluded that there are different profiles depending on the levels of personal and social responsibility during the Secondary and High School stage. In turn, these profiles indicate that students who show higher levels of responsibility are also those who show more internal motivation and greater satisfaction with basic psychological needs while generating a better school social climate and lower values of violence. It is noteworthy that these positive results occur for the profile with more responsibility regardless of age or gender, while no differences in the distribution of profiles based on any of these variables were found”. Again – please rethink one-directed conclusions. It is a mistake.
5. And my main concern regarding conclusions. As I wrote at the begging of my review, I my opinion study design and research realization are very satisfactory. Only the conclusion is missing. In the discussion part, you indicate the relations between variables. Great. But in this part, and especially in conclusions, we do not have references to the topic, which is Physical Education. So not only the relations between responsibility and different profiles is important, but responsibility in physical education classes. Please refer to the specificity of these lessons. Why are these lessons an important element of school reality? What conclusions for the practice of physical education are from your research? This is very much lacking in the current version of the text.
Author Response
Thank you to the reviewer. I include the answer in red and in the manuscript underlined with yellow color
Dear Author,
It was a great pleasure to read your article. The topic is very actual and extremely important. And – what the most important – study design and research realization are very satisfactory. My concerns are connected with the conclusions of the study.
Please consider my suggestions, because there are some points to improve.
1. Abstract – please rethink using the statement „three basic psychological needs”. It is not correct.
This statement have been changed. But the reviewer consider this change is not correct, please let me know so I can attend to it more accurately.
2. Abstract – „It is concluded that the promotion of responsibility in Physical Education classes and in the general educational field, can have positive results in obtaining a more adaptive psychological profile”. Please notice, that we can not indicate the direction of these relations. Also, the more adaptive psychological profile can have positive results in responsibility in Physical Education classes. We can only diagnose the relations.
It have been changed taking into accunt the new conclusions and new PE references.
3. Introduction – line 40: „homework?”. It must be a mistake
It have changed and the manuscript has been revised again.
4. Conclusions – „It is concluded that there are different profiles depending on the levels of personal and social responsibility during the Secondary and High School stage. In turn, these profiles indicate that students who show higher levels of responsibility are also those who show more internal motivation and greater satisfaction with basic psychological needs while generating a better school social climate and lower values of violence. It is noteworthy that these positive results occur for the profile with more responsibility regardless of age or gender, while no differences in the distribution of profiles based on any of these variables were found”. Again – please rethink one-directed conclusions. It is a mistake.
5. And my main concern regarding conclusions. As I wrote at the begging of my review, I my opinion study design and research realization are very satisfactory. Only the conclusion is missing. In the discussion part, you indicate the relations between variables. Great. But in this part, and especially in conclusions, we do not have references to the topic, which is Physical Education. So not only the relations between responsibility and different profiles is important, but responsibility in physical education classes. Please refer to the specificity of these lessons. Why are these lessons an important element of school reality? What conclusions for the practice of physical education are from your research? This is very much lacking in the current version of the text.
The bibliography has been expanded with more references in relation to physical education and the variables under study. In the same way, the discussion and the conclusions have been modified (in yellow) taking into account their considerations and seeking the link with physical education and importance to this paper
Best regards
Reviewer 3 Report
Dear Author
Thank you very much for your interesting manuscript, I think it can be highly relevant in the scientific community.
However, with a view to its possible publication, I believe that some aspects should be taken into account:
INTRODUCTION
Line 40: remove the question mark. Along these same lines, I would look for more coherence in the paragraph between references 7, 8 and. 9
Line 75, an attempt should be made to standardize the term Physical Education, sometimes PE is used and others in its entirety
In general, the framework is adequate, since I recommend to look for more research about violence in school context to include here or in discussion section if the author find more references.
Material and methods
Study design: were the reports of the students who were of legal age collected? It should be specified if the high school students all attended Physical Education by the respondents. I imagine that students aged 19-20 would be repeaters, is this so?
Procedure: How many centers participated in the study? Was it for accessibility?
Instruments: Why was the omega coefficient calculated and not the Cronbach's alpha? The CUVE scale has many subscales. Was the differentiation of the scales taken into account in the analysis in addition to the global value?
Statistical analysis: I consider that the profile analysis was carried out correctly, but was the normality of the data taken into account to subsequently carry out the MANOVA analysis?
Results
Line 222: 3.2 must be lowered one space and in the table, the meaning of R and IM must be included in the legend
Table 3: the number 2 must be placed under GL (instead of two)
Discussion
Line 285: reference 18 should have a coam after (18, 49-51)
Line 340: missing space after applied
Limitations: The socio-economic background of the participants could influence the extension of the results, so it would be included as a possible limitation.
References
33: The comma must be removed after the name of the magazine
44: put a semicolon to separate the authors
49: put point. And comma to separate the authors
57: one space left over in second author
58, 62, 63, 66, the comma after the journal name must be removed
Thank you very much for your work,
Best wishes,
Author Response
Thank you to the reviewer. I include the answer here in red and in the manuscript underlined with yellow color
Dear Author
Thank you very much for your interesting manuscript, I think it can be highly relevant in the scientific community.
However, with a view to its possible publication, I believe that some aspects should be taken into account:
INTRODUCTION
Line 40: remove the question mark. Along these same lines, I would look for more coherence in the paragraph between references 7, 8 and. 9
Line 75, an attempt should be made to standardize the term Physical Education, sometimes PE is used and others in its entirety
In general, the framework is adequate, since I recommend to look for more research about violence in school context to include here or in discussion section if the author find more references.
The bibliography has been expanded with more references in relation to physical education and the variables under study. In the same way, the discussion and the conclusions have been modified (in yellow) taking into account their considerations and seeking the topic
Thank you to the reviewer for all your comments, I consider the manuscript has improved substantially thanks to your considerations.
Material and methods
Study design: were the reports of the students who were of legal age collected? It should be specified if the high school students all attended Physical Education by the respondents. I imagine that students aged 19-20 would be repeaters, is this so?
This aspect have been specified in the material section (105-110)
Procedure: How many centers participated in the study? Was it for accessibility?
This have been included (line 110-12)
Instruments: Why was the omega coefficient calculated and not the Cronbach's alpha? The CUVE scale has many subscales. Was the differentiation of the scales taken into account in the analysis in addition to the global value?
This coefficient was selected based on research that indicates that the use of Cronbach's coefficient in sociological studies is not the most appropriate due to the characteristics of the variables, in addition to reducing limitations of the Cronbach's coefficient such as its scarcity depending on the number of responses. More information about these aspects can be found in the scientific literature, such as the article by Ventura-León et al. (https://www.redalyc.org/pdf/773/77349627039.pdf). In relation to CUVE questionnaire, the variables were not used in this study due to wide variety of variables studied following SDT theory. I
Statistical analysis: I consider that the profile analysis was carried out correctly, but was the normality of the data taken into account to subsequently carry out the MANOVA analysis?
Before to carrying out MANOVA, we check to Box’s M test and Levene’s test of homogeneity of variance tests to check some of the parametric statistical assumption.
Results
Line 222: 3.2 must be lowered one space and in the table, the meaning of R and IM must be included in the legend
Table 3: the number 2 must be placed under GL (instead of two)
Thank you to the reviewer, This mistake have been removed
Discussion
Line 285: reference 18 should have a coam after (18, 49-51)
Line 340: missing space after applied
Limitations: The socio-economic background of the participants could influence the extension of the results, so it would be included as a possible limitation.
Thank you, a new section with limitations and future lines have been included.
References
33: The comma must be removed after the name of the magazine
44: put a semicolon to separate the authors
49: put point. And comma to separate the authors
57: one space left over in second author
58, 62, 63, 66, the comma after the journal name must be removed
All references have been revised and modified
Thank you very much for your work,
Best wishes,
Thank you for all your comments to improve this manuscript.